# Surface Characteristics of Esthetic Nickel–Titanium and Beta-Titanium Orthodontic Archwires Produced by Plasma Electrolytic Oxidation (PEO)—Primary Results

**DOI:** 10.3390/ma12091403

**Published:** 2019-04-30

**Authors:** Ole Jung, Jean-Philippe Becker, Ralf Smeets, Martin Gosau, Germain Becker, Bärbel Kahl-Nieke, Anne-Kathrin Jung, Max Heiland, Alexander Kopp, Mike Barbeck, Till Koehne

**Affiliations:** 1Division of Regenerative Orofacial Medicine, Research Group Biomaterials/Surfaces, Department of Oral and Maxillofacial Surgery, University Medical Center Hamburg-Eppendorf, 20246 Hamburg, Germany; r.smeet@uke.de (R.S.); anne_1510@hotmail.com (A.-K.J.); 2Department of Oral and Maxillofacial Surgery, University Medical Center Hamburg-Eppendorf, 20246 Hamburg, Germany; m.gosau@uke.de (M.G.); becker2@me.com (G.B.); 3Department of Orthodontics, University Medical Center Hamburg-Eppendorf, 20246 Hamburg, Germany; niecke@uke.de; 4Charité-Universitätsmedizin Berlin, Corporate Member of Freie Universität Berlin, Humboldt-Universität zu Berlin and Berlin Institute of Health, Department of Oral and Maxillofacial Surgery, 12200 Berlin, Germany; max.heiland@cahrite.de; 5Meotec GmbH & Co. KG, 52068 Aachen, Germany; alexander.kopp@meotec.eu

**Keywords:** esthetic archwires, coating stability, backscattered electron microscopy

## Abstract

Background/Aim: There is continuing interest in engineering esthetic labial archwires. The aim of this study was to coat nickel–titanium (NiTi) and beta-titanium (β-Ti), also known as titanium molybdenum (TMA), archwires by plasma electrolytic oxidation (PEO) and to analyze the characteristics of the PEO-surfaces. Materials and Methods: PEO-coatings were generated on 0.014-inch NiTi and 0.19 × 0.25-inch β-Ti archwires. The surfaces were analyzed by scanning electron microscopy and stereomicroscopy. Cytocompatibility testing was performed with ceramized and untreated samples according to EN ISO 10993-5 in XTT-, BrdU- and LDH-assays. The direct cell impact was analyzed using LIVE-/DEAD-staining. In addition, the archwires were inserted in an orthodontic model and photographs were taken before and after insertion. Results: The PEO coatings were 15 to 20 µm thick with a whitish appearance. The cytocompatibility analysis revealed good cytocompatibility results for both ceramized NiTi and β-Ti archwires. In the direct cell tests, the ceramized samples showed improved compatibility as compared to those of uncoated samples. However, bending of the archwires resulted in loss of the PEO-surfaces. Nevertheless, it was possible to insert the β-Ti PEO-coated archwire in an orthodontic model without loss of the PEO-ceramic. Conclusion: PEO is a promising technique for the generation of esthetic orthodontic archwires. Since the PEO-coating does not resist bending, its clinical use seems to be limited so far to orthodontic techniques using straight or pre-bent archwires.

## 1. Introduction

Labial brackets and wires are rejected by a large proportion of adult patients and increasingly by adolescent patients. This led to the development of more esthetic orthodontic appliances such as lingual fixed brackets or clear plastic aligners. However, these devices have certain drawbacks in terms of costs, ease of use and effectiveness [1,2,3]. There is, therefore, a continuing need for labial fixed appliances.

The development of tooth-colored ceramic brackets (e.g., polyoxymethylene, glass fiber, sapphire) has significantly improved the esthetics and acceptance of labial fixed appliances [4,5]. Attempts have also been made to manufacture esthetic orthodontic archwires by coating with polymer or Teflon. However, these coatings proofed to have a poor wear resistance, which resulted in unfavorable esthetic and mechanical properties after a short time of treatment [6,7]. New esthetic materials such as fiber-reinforced plastic wires have been recently introduced in dentistry because of their promising strength and stiffness per weight when compared with other structural materials along with adequate toughness [8]. However, fiber-reinforced plastic wires were clinically unsuccessful due to their overall inferior mechanical properties when compared to those of conventional nickel-titanium (NiTi) and beta-titanium (β-Ti) archwires [9]. 

Plasma electrolytic oxidation (PEO) is a novel treatment method to ceramized metallic alloys [10,11]. In contrast to conventional surface treatments, the ceramic layer is not a separate layer on the substrate, but rather an inert part of the ceramized alloy. As a result, the mechanical properties like hardness and abrasion are sustainably improved [12,13]. By means of specific adjustments and electrolytic compositions, it is also possible to produce an esthetic white appearance of the ceramic layer. 

The aim of this study was therefore to employ PEO for the production of esthetic orthodontic archwires. For this purpose, Niti and β-Ti wires were ceramized, analyzed in vitro and inserted in an ex vivo model. 

## 2. Materials and Methods

### 2.1. Biomaterial Preparation and Physiochemical Analysis

Two types of orthodontic archwires were used for this study: 0.014-inch NiTi (NiTi classic, 3M, Neuss, Germany) and 0.19 × 0.25-inch β-Ti (TMA, Ormco, Amersfoort, Netherlands) archwires. The 0.014-inch NiTi archwire is a highly flexible wire that is used for the initial alignment of irregular teeth [14]. The 0.19 × 0.25-inch β-Ti archwire is a more rigid archwire that is used for the working and final phase of orthodontic treatment [15,16]. These two archwires cover therefore most of the treatment tasks during fixed orthodontic therapy.

The archwires were ceramized using plasma electrolytic oxidation (PEO). PEO was achieved using a pulsed rectifier set (Meotec M-PEO A1, Aachen, Germany). Positive and negative pulsed galvanostatic currents of 2 A and 3 A with voltages from 0 V to 500 V were applied using two feed cables. The electrochemical cell consisted of a cool-jacketed glass vessel containing the specimens to be coated as anode and a platinum sheet as cathode. Pulse frequency was set to 20 Hz and discs were treated for up to 15 minutes. After processing, all samples were rinsed with distilled water in an ultrasonic bath for 15 minutes and dried on a sterile blanket at room temperature.

The coated archwires were scanned with a scanning electron microscope (XL30 CP SEM, Philips, Amsterdam, The Netherlands) at 20 kV and a 680-pA electron beam current. The stereomicroscopic images were obtained with a high-resolution inverted microscope (Olympus DSX500i, Olympus Europa SE & CO. KG, Hamburg, Germany) and a DSX-10 × objective lens (Leica Microsystems GmbH, Wetzlar, Germany). The archwires were bent under visual control in the microscope till the coating chipped off. 

### 2.2. Cytocompatibility Analyses

Cytocompatibility analyses were conducted in accordance with the ISO 10993-5/-12 and were already described in detail previously [13,17]. Overall, tests can be divided into indirect (extract) and direct (live-dead analysis) processes in which colorimetric assays and fluorescence stains of cells give a feedback on sample toxicity [13,17,18,19]. In brief, L-929 mouse fibroblasts (European Collection of Cell Culture, Salisbury, UK) were used to test ceramized and untreated NiTi and β-Ti archwires in comparison with positive (RM-A, Hatano Research Institute, Food and Drug Safety Center, Ochiai, Japan) and negative control samples (Wako Pure Chemical Industries, Ltd., Osaka, Japan, Cat. No.160-08893). 

For extraction analysis, all test samples were extracted for 72 hours in cell culture medium (MEM (Minimum Essential Medium), 10% fetal bovine serum, penicillin/streptomycin (100 U/mL each, all from Life Technologies, Carlsbad, USA) and 4mM L-glutamine (Sigma-Aldrich, St. Louis, USA) under cell culture conditions (37 °C, 5% CO_2_ and 95% humidity) with a surface to volume ratio of 3 cm^2^/mL. We chose 72 hours for a better and more rigorous analysis of the test samples. Viability, proliferation and cytotoxicity were determined using XTT-, BrdU (both: Roche Diagnostics, Mannheim, Germany) and LDH (BioVision, Milpitas, CA 95035, USA) assays according to the manufacturer’s instructions.

For live–dead analysis, all test samples were seeded with 2.4 × 10^5^ cells in 1 mL cell culture medium in 12 well plates (surface-area/medium ratio: 5.65 cm^2^/mL). Assays were carried out after 24 h incubation using 60 µL per mL medium propidium iodide (PI) stock solution (50 µg/mL in PBS) and 500 µL per mL medium fresh fluorescein diacetate (FDA) working solution. Incubation time was set for 3 minutes at room temperature. Thereafter, specimens were rinsed in prewarmed PBS and immediately examined using an upright fluorescence microscope (Nikon ECLIPSE Ti-S/L100, Nikon GmbH, Düsseldorf, Germany). Pictures were taken using a 4×, 10× and 20× objective.

### 2.3. Ex Vivo Analyses

To get a first impression of clinical usability, the ceramized NiTi and β-Ti archwires were inserted in an orthodontic plastic model with ceramic brackets (Clarity^TM^, 3M, Neuss, Germany). The insertion and ligation of the archwire were repeated three times, mimicking a practical point of view, using the same archwire. Photographs were taken with a digital camera before and after insertion. 

### 2.4. Statistical Analyses

The data of the cytocompatibility extract results were statistically analyzed by an analysis of variance (ANOVA) combined with a Tukey’s multiple comparisons test via the GraphPad Prism 7.0d software (GraphPad Software Inc., La Jolla, USA). Normal distribution was analyzed beforehand by Kolmogorov–Smirnov. Statistical significant differences were designated as follows: significant different if p-values were less than 0.05 (* p ≤ 0.05) and highly significant if P-values were less than 0.01 (** p ≤ 0.01) or less than 0.001 (*** p ≤ 0.001). 

## 3. Results

### 3.1. Material Analyses

Both NiTi and β-Ti archwires could be successfully ceramized using PEO. Macroscopically, a whitish tooth-like colored surface could be generated. SEM revealed a uniform PEO-layer on the NiTi and β-Ti archwires (Figure 1). The PEO-particles were deposited along the entire surfaces and formed an approximate layer between 20 to 30 µm in thickness. The surface of the PEO-layer had a higher degree of micro-roughness as compared to those of non-coated wires. 

### 3.2. Cytocompatibility

Cytocompatibility analyses revealed that both treated and untreated NiTi and β-Ti archwires are safe according EN ISO 10993-5/-12 [20,21].

In the extract tests, values <130% of the negative control in the cytotoxicity assay and values >70% of the negative control in the viability and proliferation assays are within the nontoxic range. In all assays, the values of the positive control were significantly different compared to the other test samples (p ≤ 0.001) (Figure 2). In the XTT assay, the experimental group NiTi-PEO revealed significant differences compared to both untreated archwires (p ≤ 0.05). Both ceramized and untreated archwires showed no significant differences compared to the negative control. 

In the direct tests, all test materials and the negative control showed mostly green living and adherent cells compared to the positive control, on which fewer and red dead cells were visible (Figure 3). Interestingly, a few red dead cells were visible on the untreated NiTi and β-Ti, while both ceramized pendants were comparable with the negative control.

### 3.3. Ex Vivo Analysis

The stereomicroscopic examination showed PEO-coatings along the entire surface of both archwires (Figure 4). The grey color of the NiTi and β-Ti archwires gleamed slightly through the PEO-coatings. Deformation of the archwires resulted in cracks in the PEO-layers and peeling of the PEO-coatings.

Finally, PEO-coated and non-coated archwires were inserted in ceramic brackets in an orthodontic phantom model and ligated with orthodontic elastics (Figure 5). However, the PEO-coating on the NiTi archwire showed poor wear resistance and approximately half of the coating was lost after multiple insertions. In contrast, the PEO-coating on the β-Ti archwire was stable also after three times of insertion and ligation. Importantly, the PEO-coatings had a pleasant tooth-colored appearance.

This procedure was repeated three times using the same archwire. Both archwires could be easily inserted in the slots of the ceramic brackets and the clinical handling of PEO-coated archwires was not different from that of non-coated control wires.

## 4. Discussion

Our results indicate that PEO is a promising technique for the production of esthetic whitish orthodontic archwires. Thereby, PEO-archwires show sufficient in vitro cytocompatibility in comparison with untreated wires. In terms of biomechanical properties, only PEO of β-Ti wires can be classified suitable for further investigations and patient treatment. We could not investigate stainless steel archwires since PEO is not applicable on this material. Overall, there is a clear need for more studies in terms of biomechanical properties and wear resistance. 

In fact, one concern of the PEO-coated wires for orthodontic use could be the rigidity of the ceramic layer. We observed that the deformation of the archwires resulted in cracks and chipping of the PEO-coating. This may also explain the high degree of coating loss of the NiTi wires after insertion into the brackets. Round NiTi archwires are highly flexible and deformed during insertion since they are used for the initial alignment of irregular teeth. For the clinical use of PEO-coated round NiTI archwires it would be therefore necessary to substantially increase the flexibility of the PEO coating. This would also improve the resistance of the PEO-coatings on the thicker β-Ti archwires. In fact, our stereomicroscopic analysis demonstrated that bending of the β-Ti archwire also resulted in loss of the PEO-coating. This can counteract the esthetic reliability and overall functionality. Furthermore, sharp-edged corrosion particles could harm teeth and soft tissue of the digestive tract. However, this may be clinically less critical since these archwire are less deformed during insertion. Accordingly, we observed that PEO-coated β-Ti archwires showed a good resistance to multiple insertion and ligation into the brackets. It would therefore be possible to use these wires for orthodontic technique using straight wires without bending. Moreover, it is important to mention, that nowadays several customized labial appliances systems are on the market, which are using fully individualized pre-bent wires [22]. PEO-coatings of these pre-bent wires could substantially improve the patients’ acceptance of these appliances. It is also important, that PEO-coated surfaces have generally a high resistance to mechanical wearing due to its ceramic characteristics [13]. In fact, a previous study demonstrated that PEO might even increase the wear resistance of NiTi alloys [23]. It is therefore likely, that PEO-coated archwires have a higher resistance against abrasion by food or tooth brushing as compared to conventional coated wires. This clearly needs to be tested in future studies. 

Comparable with other studies, PEO shows an overall sufficient cytocompatibility [11,17,24,25,26]. Thereby, NiTi shows slightly lower values in the extract tests compared to the untreated test samples and β-Ti. However, these values were not within the toxic range according to EN ISO 10993-5/-12 and may occur as test variability [13,17]. The sufficient attachment of cells on the ceramic surface can be explained the rough ceramic surfaces of the test samples [11]. As with the conventional surfaces, food residues and other deposits can easily occur and could be avoided. The development of non-adherent ceramic surfaces could be an interesting approach for further research efforts. Furthermore, as shown in other studies of our group, PEO can be universally applied on different alloys like titanium and magnesium, but not steel [11,13,17]. Thereby, similar to the present study, PEO can increase the compatibility of biomaterials [11,13,17].

There is also a definite need to characterize the mechanical properties of the PEO-coated archwires. These studies should include conventional bending tests and the characterization of resistance to friction. In fact, it remains to be tested, whether the micro-roughness of the PEO-coating increases the friction between the archwire and brackets. It is important to address this question not only in comparison to uncoated archwires, but also in comparison to conventional coatings, which show a degree of roughness in clinical use.

Taken together, this proof-of-concept study demonstrates that PEO is a feasible technique for the production of esthetic whitish orthodontic archwires. PEO seems to be most promising so far for orthodontic labial appliances using straight or pre-bent archwires. However, there is a clear need for further studies characterizing the biomechanical properties (e.g., friction, wear resistance) of the PEO-coated archwires. Furthermore, PEO technology should be extended by the ceramization of steel. 

## Figures and Tables

**Figure 1 materials-12-01403-f001:**
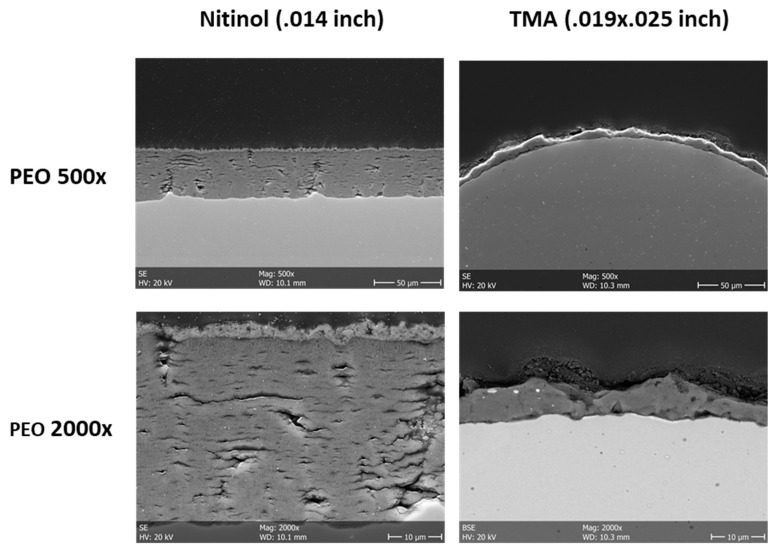
Scanning electron microscopy of non-coated and plasma electrolytic oxidation (PEO)-coated NiTi and β-Ti archwires. The PEO-layer has a sufficient thickness and a certain degree of micro-roughness (left panels: cross-sections, right panels: top views).

**Figure 2 materials-12-01403-f002:**
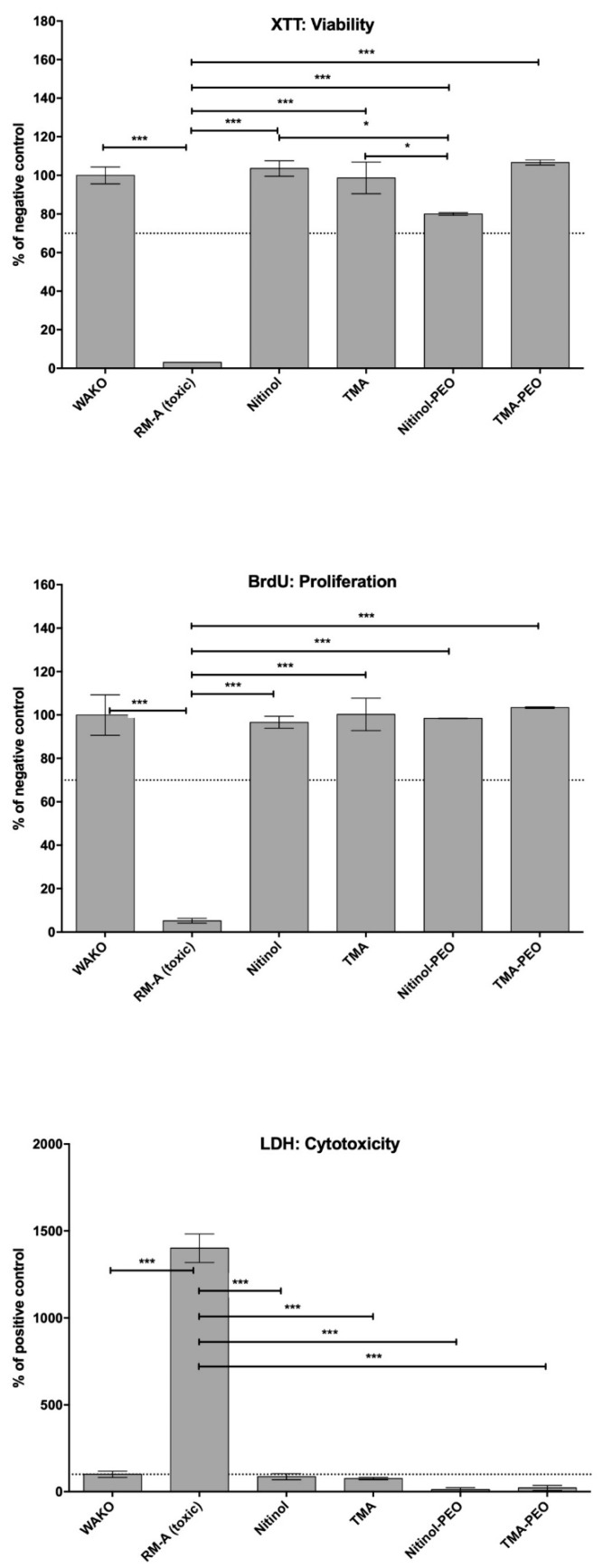
Cytocompatibility extract results achieved by XTT, BrdU and LDH assays. Statistical significant differences are as follows: * p ≤ 0.05 (significant different), ** p ≤ 0.01 or *** p ≤ 0.001 (high significant difference).

**Figure 3 materials-12-01403-f003:**
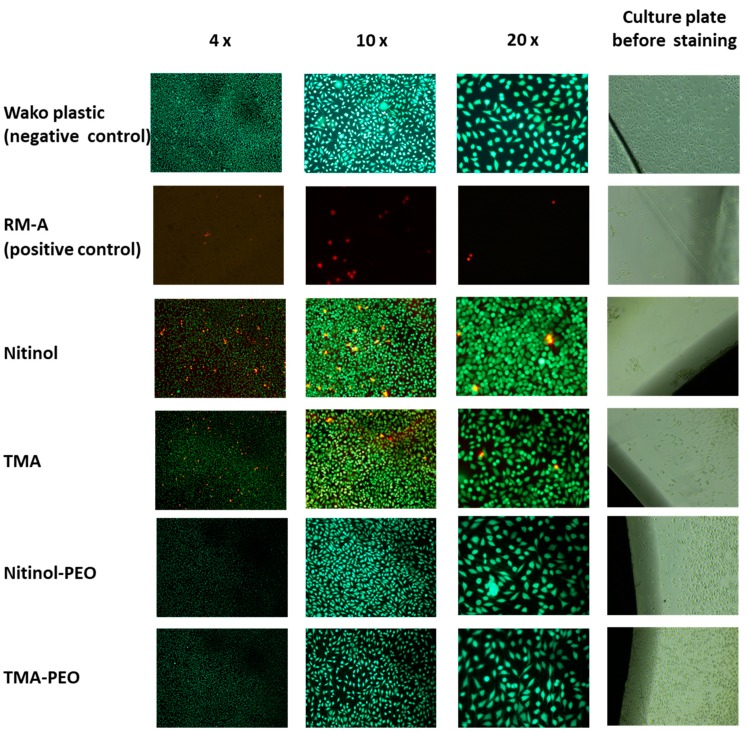
Live–dead staining of both ceramized and untreated NiTi and β-Ti archwires compared to RM-A positive control and WAKO negative control.

**Figure 4 materials-12-01403-f004:**
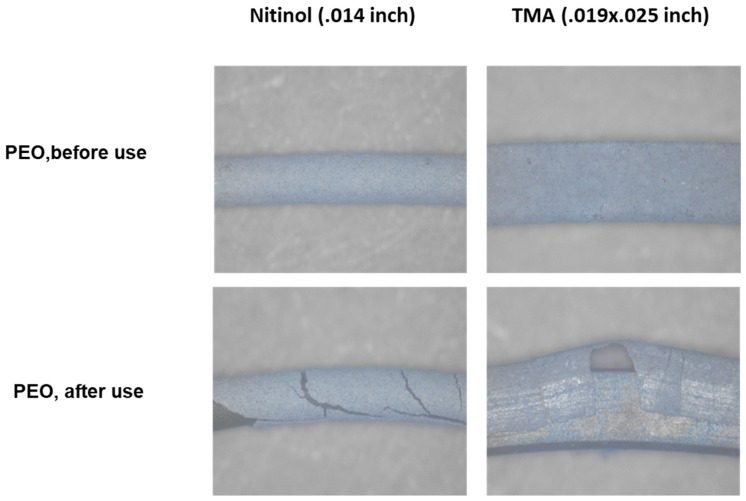
Stereomicroscopy of PEO-coated NiTi and β-Ti archwires. Lower panels show loss of PEO-coating after bending of the archwires.

**Figure 5 materials-12-01403-f005:**
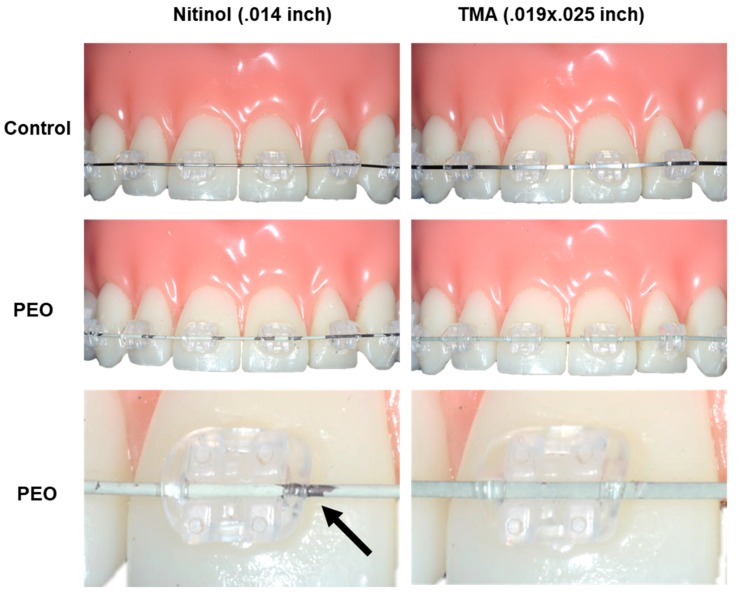
PEO-coated NiTi and β-Ti archwires in an orthodontic phantom model. Non-coated archwires are shown for comparison (upper panels). Multiple insertion of the NiTi archwires results in loss (black arrow) of the PEO-coating (middle and lower panels). In contrast, The PEO-coating on the β-Ti wire is stable also after repeated insertion.

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
