# Peer review of "Surface Characteristics of Esthetic Nickel–Titanium and Beta-Titanium Orthodontic Archwires Produced by Plasma Electrolytic Oxidation (PEO)—Primary Results"

_materials, 2019, doi:10.3390/ma12091403_

Reviewer 1 Report

Dear Authors,

I have read the manuscript and I have found it interesting for orthodontists. Some questions raised. Enlisted please find my comments.

Abstract. Line 24. “Cytocombability”. Please      correct into “Cytocompatibility”.

Abstract. Line 28. Authors stated that “The PEO coatings      were 15 to 20 μm thick and esthetically pleasing.”. Esthetical      consideration is strictly subjective. Moreover, esthetic evaluation was      not among the scopes of the present research. Please remove “esthetically      pleasing”.

Introduction. Authors stated “The development of      tooth-colored ceramic brackets has significantly improved the aesthetics      and acceptance of labial fixed appliances (4)”. It is true but incomplete.      Aesthetic brackets realized with other materials have been recently      introduced. Authors could state that “The development of tooth-colored      ceramic, polyoxymethylene, glass fiber, sapphire, brackets (Light energy      transmission through six different makes of ceramic orthodontic brackets. Aldossary      MS, Abu Hajia SS, Santini A. Int Orthod. 2018 Dec;16(4):638-651) has      significantly improved the aesthetics and acceptance of labial fixed      appliances (4)”.

Introduction. Authors stated “New esthetic      materials such as fiber-reinforced plastic wires were also clinically      unsuccessful due the inferior mechanical properties of these archwires as      compared to those of conventional nickel-titanium (Niti) and beta-titanium      (Beta-Ti) archwires (7).”. The sentence could be explained more      extensively. Authors could state that “New esthetic materials such as      fiber-reinforced plastic have been recently introduced in Dentistry with      many purposes. These materials present high stiffness and strength per      weight when compared with other structural materials along with adequate      toughness (Fiber-Reinforced Composites for Dental Applications. Scribante      A, Vallittu PK, Özcan M. Biomed Res Int. 2018 Nov 1;2018:4734986.).      However, fiber reinforced plastic orthodontic wires have been tested but      they were clinically unsuccessful due the inferior mechanical properties      of these archwires as compared to those of conventional nickel-titanium      (Niti) and beta-titanium (Beta-Ti) archwires (7).”

Materials and Methods. Authors stated “0.014-inch      NiTi archwire is a highly flexible wire that is used for the initial      alignment of irregular teeth”. Please add a reference for this statement.

Materials and Methods. Authors stated “The      0.19x0.25-inch Beta-Ti archwire is a more rigid archwire that is used for      the working and final phase of orthodontic treatment”. Please add a      reference for this statement.

Materials and Methods. Authors stated “For      extraction analysis, all test samples were extracted for 72 hours in cell      culture medium (MEM (Minimum Essential Medium), 10% fetal bovine serum,      penicillin/streptomycin (100 U/mL each, all from Life Technologies,      Carlsbad, USA) and 4mM L-glutamine (Sigma-Aldrich, St. Louis, USA) under      cell culture conditions (37 °C, 5% CO2 and 95% humidity) with a surface to      volume ratio of 3 cm2/ml.”. Please add a reference for this method.

Materials and Methods. Authors stated “The      insertion and ligation of the archwire were repeated three times using the      same archwire”. Why Authors decided three times? Please specify if this      method has been used also in previous studies and if a sample size      calculation has been performed

Materials and Methods. Authors stated “The data      of the cytocompatibility extract results were statistically analyzed by an      analysis of variance (ANOVA) combined with a Tukey’s multiple comparisons”.      ANOVA is used only for data that follow Gaussian distribution. Please      state if data distribution has been tested before choosing ANOVA and      specify the name of the test (e.g.: Kolmogorov-Smirnov?).

Materials and Methods. Authors stated “Macroscopically,      an aesthetically pleasing surface could be generated”. Aesthetically      pleasing surface is subjective. Please change the sentence into      “tooth-like surface” or somewhat else.

Materials and Methods. Authors stated “The      PEO-particles were deposited along the entire surfaces and formed a layer      between 15 to 20 μm in thickness”. Please add a reference for this      statement

Results. Authors reported that “Cytocompatibility      analyses revealed that both treated and untreated NiTi and Beta-Ti      archwires are safe according EN ISO 10993-5/-12.”. Please add the      reference of the ISO standard.

Results. Authors reported that “Both ceramized      and untreated archwires showed no significant differences compared to the      negative control.”. Please add P value.

Results. Authors reported that “all test      materials and the negative control showed mostly green living and adherent      cells compared to the positive control, on which fewer and red dead cells      were visible”. Please add P value.

Results. Authors reported that “both ceramized      pendants revealed superior compatibility comparable with the negative      control”. Please add P value.

Results. Authors stated “Both archwires could be      easily inserted in the slots of the ceramic brackets and the clinical      handling of PEO-coated archwires was not different from that of non-coated      control wires”. These are personal considerations that are not related to      the results of the study and should be cropped or moved into discussion      section and discussed further with proper references.

Discussion. The paragraph is very short. Authors      could elongate it pointing out some more aspects related to their      experiments.

Discussion. Authors tested niti and TMA wires. In      fact, many orthodontists use stainless steel wires during final movements.      Therefore, a sentence about the lack of testing stainless steel wires      should be added.

Discussion. Authors performed the insertion and      ligation of the archwire were repeated three times using the same archwire.      Please discuss (as a limitation of the present report) the importance of      wear that could alter the results.

Discussion. PEO coated wires seems to be a viable      alternative to conventional wires. However this new technology has to be      tested further before clinical use: friction and mechanical behavior of      the wires should be tested in future. This should be pointed out at the      end of discussion section.

References. Few references have been reported.      Please add some more studies. Some references have been suggested in      previous comments.

Figures. In the version that I have downloaded I      cannot see Figures. Please resubmit with figures added in the main text.

Tables. No tables have been given showing the      results. Please add a table with descriptive statistics (Mean, Standard      deviation, Minimum, Median and Maximum for each group tested) of the      results.

Author Response

Reviewer 1

Abstract. Line 24. “Cytocombability”. Please      correct into “Cytocompatibility”.

Answer: Done.

Abstract. Line 28. Authors stated that “The PEO coatings      were 15 to 20 μm thick and esthetically pleasing.”. Esthetical      consideration is strictly subjective. Moreover, esthetic evaluation was      not among the scopes of the present research. Please remove “esthetically      pleasing”.

Answer: Done.

Introduction. Authors stated “The development of      tooth-colored ceramic brackets has significantly improved the aesthetics      and acceptance of labial fixed appliances (4)”. It is true but incomplete.      Aesthetic brackets realized with other materials have been recently      introduced. Authors could state that “The development of tooth-colored      ceramic, polyoxymethylene, glass fiber, sapphire, brackets (Light energy      transmission through six different makes of ceramic orthodontic brackets. Aldossary      MS, Abu Hajia SS, Santini A. Int Orthod. 2018 Dec;16(4):638-651) has      significantly improved the aesthetics and acceptance of labial fixed      appliances (4)”.

Answer: We added the suggested improvements and literature.

Introduction. Authors stated “New esthetic      materials such as fiber-reinforced plastic wires were also clinically      unsuccessful due the inferior mechanical properties of these archwires as      compared to those of conventional nickel-titanium (Niti) and beta-titanium      (Beta-Ti) archwires (7).”. The sentence could be explained more      extensively. Authors could state that “New esthetic materials such as      fiber-reinforced plastic have been recently introduced in Dentistry with      many purposes. These materials present high stiffness and strength per      weight when compared with other structural materials along with adequate      toughness (Fiber-Reinforced Composites for Dental Applications. Scribante      A, Vallittu PK, Özcan M. Biomed Res Int. 2018 Nov 1;2018:4734986.).      However, fiber reinforced plastic orthodontic wires have been tested but      they were clinically unsuccessful due the inferior mechanical properties      of these archwires as compared to those of conventional nickel-titanium      (Niti) and beta-titanium (Beta-Ti) archwires (7).”

Answer: We added the suggested improvements and literature.

Materials and Methods. Authors stated “0.014-inch      NiTi archwire is a highly flexible wire that is used for the initial      alignment of irregular teeth”. Please add a reference for this statement.

Answer: Done.

Materials and Methods. Authors stated “The      0.19x0.25-inch Beta-Ti archwire is a more rigid archwire that is used for      the working and final phase of orthodontic treatment”. Please add a      reference for this statement.

Answer: Done.

Materials and Methods. Authors stated “For      extraction analysis, all test samples were extracted for 72 hours in cell      culture medium (MEM (Minimum Essential Medium), 10% fetal bovine serum,      penicillin/streptomycin (100 U/mL each, all from Life Technologies,      Carlsbad, USA) and 4mM L-glutamine (Sigma-Aldrich, St. Louis, USA) under      cell culture conditions (37 °C, 5% CO2 and 95% humidity) with a surface to      volume ratio of 3 cm2/ml.”. Please add a reference for this method.

Answer: We cited two references from our previous research at the beginning of the paragraph. If necessary, we could cite both references after each sentence but are afraid that the clarity could be impaired.

Materials and Methods. Authors stated “The      insertion and ligation of the archwire were repeated three times using the      same archwire”. Why Authors decided three times? Please specify if this      method has been used also in previous studies and if a sample size      calculation has been performed

Answer: This part of study was performed from a practical point. We repeated the insertion three times because we think that materials should be reliable by this number in orthodontics. We tried to improve the sentence by writing that we decided to repeat the insertion three times because from our practical point of view. We cannot provide any appropriate literature.

Materials and Methods. Authors stated “The data      of the cytocompatibility extract results were statistically analyzed by an      analysis of variance (ANOVA) combined with a Tukey’s multiple comparisons”.      ANOVA is used only for data that follow Gaussian distribution. Please      state if data distribution has been tested before choosing ANOVA and      specify the name of the test (e.g.: Kolmogorov-Smirnov?).

Answer: Done.

Materials and Methods. Authors stated “Macroscopically,      an aesthetically pleasing surface could be generated”. Aesthetically      pleasing surface is subjective. Please change the sentence into      “tooth-like surface” or somewhat else.

Answer: We improved this sentence.

Materials and Methods. Authors stated “The      PEO-particles were deposited along the entire surfaces and formed a layer      between 15 to 20 μm in thickness”. Please add a reference for this      statement

Answer: We analyzed all our SEM pictures and measured the approximate thickness of the ceramic. We revised this sentence and wrote “20 to 30 µm in thickness”

Results. Authors reported that “Cytocompatibility      analyses revealed that both treated and untreated NiTi and Beta-Ti      archwires are safe according EN ISO 10993-5/-12.”. Please add the      reference of the ISO standard.

Answer: Done.

Results. Authors reported that “Both ceramized      and untreated archwires showed no significant differences compared to the      negative control.”. Please add P value.

Answer: There are many p-values since of the multiple comparisons. We ask not to show these values in detail to keep overall clarity.

Results. Authors reported that “all test      materials and the negative control showed mostly green living and adherent      cells compared to the positive control, on which fewer and red dead cells      were visible”. Please add P value.

Results. Authors reported that “both ceramized      pendants revealed superior compatibility comparable with the negative      control”. Please add P value.

Answer: We revised the sentences. We cannot add a p-value because the direct evaluation compares all test samples from a qualitative point of view.

Results. Authors stated “Both archwires could be      easily inserted in the slots of the ceramic brackets and the clinical      handling of PEO-coated archwires was not different from that of non-coated      control wires”. These are personal considerations that are not related to      the results of the study and should be cropped or moved into discussion      section and discussed further with proper references.

Answer: We revised this part and further discussed the practical considerations in the discussion section.

Discussion. The paragraph is very short. Authors      could elongate it pointing out some more aspects related to their      experiments.

Answer: We revised the discussion and tried to insert some interesting points.

Discussion. Authors tested niti and TMA wires. In      fact, many orthodontists use stainless steel wires during final movements.      Therefore, a sentence about the lack of testing stainless steel wires      should be added.

Answer: To our knowledge and experiments, stainless steel can not be treated by means of PEO. We added a sentence that stainless steel could not be investigated and more research should be done to make PEO on steel applicable.

Discussion. Authors performed the insertion and ligation of the archwire were repeated three times using the same archwire. Please discuss (as a limitation of the present report) the importance of wear that could alter the results.

Answer: Thank you for this comment. We revised this part of the discussion and added two sentences regarding wear and corrosion particles.

Discussion. PEO coated wires seems to be a viable alternative to conventional wires. However this new technology has to be tested further before clinical use: friction and mechanical behavior of      the wires should be tested in future. This should be pointed out at the end of discussion section.

Answer: We already stated that biomechanical properties should be further investigated and revised this sentence by adding friction and wear resistance in brackets.

References. Few references have been reported.      Please add some more studies. Some references have been suggested in      previous comments.

Answer: We added some more references and extended the discussion.

Figures. In the version that I have downloaded I      cannot see Figures. Please resubmit with figures added in the main text.

Answer: We received a message of Ms. Bo beforehand and propounded a revised version with figures. We now submit a revised version with all reviewer comments and figures inside the text version.

Tables. No tables have been given showing the      results. Please add a table with descriptive statistics (Mean, Standard      deviation, Minimum, Median and Maximum for each group tested) of the      results.

Answer: We now submit a version with figures. Figure 2 shows the significant differences between the groups. In our opinion, a table with all differences may be confusing to the reader. However, if wanted, we can provide a table with all the necessary information.

Reviewer 2 Report

This is an interesting study about new oxidation process to orthodontic archwires

Some criticisms are present:

1)The citotoxicity assay should be explained better, beginning to the principal in vitro tests evaluation   to assess it.In this context these two literature works should be added and discussed in the text

Pagano S, Chieruzzi M, Balloni S, Lombardo G, Torre L, Bodo M, Cianetti S, Marinucci L Biological, thermal and mechanical characterization of modified glass ionomer cements: the role of nanohydroxyapatite, ciprofloxacin and zinc l-carnosine. Mater Sci Eng C Mater Biol Apple 2019 Jan 1;94:76-85

Balloni S, Locci P, Lumare A, Marinucci L Cytotoxicity of three commercial mouthiness on extracellular matrix metabolism and human gingival cell behavior Toxicology in vitro, Vol 34, pag 88-96, 2016

2)Auhtors shlud explain time choice of cytotoxicity test and why not more long time (24, 48, 72 h.etc.)

3)Some technical test should be added about traditional ceramic arch wires with the same culture medium

Author Response

1)The citotoxicity assay should be explained better, beginning to the principal in vitro tests evaluation   to assess it. In this context these two literature works should be added and discussed in the text

Pagano S, Chieruzzi M, Balloni S, Lombardo G, Torre L, Bodo M, Cianetti S, Marinucci L Biological, thermal and mechanical characterization of modified glass ionomer cements: the role of nanohydroxyapatite, ciprofloxacin and zinc l-carnosine. Mater Sci Eng C Mater Biol Apple 2019 Jan 1;94:76-85

Balloni S, Locci P, Lumare A, Marinucci L Cytotoxicity of three commercial mouthiness on extracellular matrix metabolism and human gingival cell behavior Toxicology in vitro, Vol 34, pag 88-96, 2016

Answer: We revised this part, added the references and give more details for the general understanding of the principle. We refer to a detailed description in other publications.

2)Auhtors shlud explain time choice of cytotoxicity test and why not more long time (24, 48, 72 h.etc.)

Answer: Done.

3)Some technical test should be added about traditional ceramic arch wires with the same culture medium

Answer: We compared the untreated test samples with the ceramized test samples (Figures 2, 3). To our knowledge, ceramic arches are not available in orthodontics.

Reviewer 3 Report

Dear Authors 

Please check the English and edit a few latest papers related to the experiment. 

Author Response

Reviewer 3:

Please check the English and edit a few latest papers related to the experiment.

Answer: We thank the reviewer for this comment and checked the manuscript accordingly.

Round  2

Reviewer 1 Report

Manuscript has been revised.

Good job!